# Systematic revision of *Gatesona* (Crassiclitellata, Lumbricidae), an endemic earthworm genus from the Massif Central (France)

**Daniel F. Marchán**[1]*, **Sergio Jiménez**[2], **Thibaud Decaëns**[1], **Jorge Domínguez**[3]

**1** CEFE, UMR 5175, CNRS–Univ Montpellier–Univ Paul–Valéry–EPHE–SupAgro Montpellier–INRA–IRD, Montpellier, France, **2** Faculty of Biology, Department of Biodiversity, Ecology and Evolution, Universidad Complutense de Madrid, Madrid, Spain, **3** Grupo de Ecoloxía Animal (GEA), Universidade de Vigo, Vigo, Spain

* daniel.fernandez-marchan@umontpellier.fr

**Data Availability Statement:** All sequence files are available in Genbank, under accession numbers

## Abstract

The Massif Central in France could potentially harbor numerous ancient endemic lineages owing to its long history of continuous geological stability. Several endemic earthworm species inhabit the area, with *Allolobophora (Gatesona) chaetophora*, *Helodrilus (Acystodrilus)* and *Avelona ligra* showing hints of a common evolutionary origin. However, the phylogenetic relationships and taxonomic status of the species remain to be studied through integrative molecular and morphological methods. To this end, eight species including most of the known species and subspecies of *All. (Gatesona)*, *Helodrilus (Acystodrilus) musicus*, and *Avelona ligra* were sequenced for a set of five molecular markers. The species were grouped on the basis of the molecular findings in a phylogenetic framework. *All. (Gatesona)* was included within the same clade as *Helodrilus (Acystodrilus)* and *Avelona*, separated from *Allolobophora* sensu stricto, supporting its status as a good genus. Branch lengths and average pairwise genetic distances suggested the subspecies of *All. (Gatesona) chaetophora* examined should be considered species-level taxa. Thus, a generic diagnosis for *Gatesona* stat. nov. is provided, along with redescriptions of *Gatesona chaetophora* comb. nov., *Gatesona rutena* comb. nov. stat. nov., *Gatesona lablacherensis* comb. nov. stat. nov. and *Gatesona serninensis* comb. nov. stat. nov. The study findings highlight the need for further sampling of earthworm diversity in the Massif Central (and Southern France), in addition to an increased focus on the Eastern European species of *Helodrilus*.

## Introduction

The Massif Central in France is an ancient massif which originated in the Variscan orogeny (Late Paleozoic) and has been an emerged continental realm through the Mesozoic and Cenozoic to the present [1]. Owing to this stability over geological time, the region is expected to harbour a large number of ancient endemic lineages (as observed for mite harvestmen in the

MZ153163-MZ153169, MZ156573-MZ156588 and
MZ235293-MZ235314.

**Funding:** This work was supported by a grant from
the Systematic Research Fund and by Xunta de
Galicia. Consellería de Cultura, Educación e
Ordenación Universitaria. Secretaria Xeral de
Universidades under grant ED431B 2019/038. DFM
was funded by a Juan de La Cierva-Formación
grant (FJCI-2017-32895) from the Spanish
Ministry of Sciences, Innovation and Universities.

**Competing interests:** The authors have declared
that no competing interests exist.

Iberian Peninsula [2]. Earthworms, whose phylogeography and biogeography are known to reflect paleogeographic events [3], are a good example of this prediction. The Massif Central is inhabited by several endemic earthworm species from narrowly distributed genera: *Hemigastrodrilus monicae* Bouché 1970 (a basal lineage of Hormogastridae [4]), *Ethnodrilus aveli* Bouché 1972, *Ethnodrilus gatesi* Bouché 1972, *Zophoscolex graffi* Bouché 1972, *Zophoscolex micellus* Bouché 1972, *Allolobophora (Gatesona) chaetophora* Bouché 1972, *Helodrilus (Acystodrilus) cortezi* Qiu & Bouché 1998, *Helodrilus (Acystodrilus) musicus* Qiu & Bouché 1998, *Helodrilus (Acystodrilus) segalensis* Qiu & Bouché 1998 and *Avelona ligra* Bouché 1969.

Among the above-mentioned species, *All. (Ga.) chaetophora*, *Helodrilus (Acystodrilus)* and *Avelona ligra* may share a common evolutionary origin according to different sources of evidence. *All. (Ga.) chaetophora* and *He. (Ac.) cortezi* were grouped together in a relatively basal clade in the molecular phylogenetics analysis reported by Domínguez et al. [5]. This position is congruent with their morphological characteristics, as excepting by their nephridial bladder structure (sigmoid vs absent) both taxa are very similar. *Av. ligra* is also endemic to the (northeastern) Massif Central and morphologically similar to both taxa. The only molecular phylogenetics study including *Av. ligra* [6] found that it is closely related to *All. (Ga.) chaetophora*; however, the sparse taxon sampling of Lumbricidae does not provide robust support for such a clade. In order to confirm a clade comprising these species, additional representatives of these taxa with a common molecular marker set must be examined.

In addition to the phylogenetic uncertainty, there remain some taxonomic questions regarding *All. (Ga.) chaetophora*. Bouché (1972) described several subspecies within *All. (Ga.) chaetophora* with very restricted ranges and significant morphological differences from the type species. Bouché acknowledged that such morphological heterogeneity should be delimited in further studies (which he was unable to perform). Surprisingly, this type of study was performed in Qiu and Bouché [7] for other species complexes such as *Scherotheca* -resulting in the elevation of several subspecies to species status- but not for *All. (Ga.) chaetophora*. However, two closely related species, *All. (Ga.) sausseyi* Qiu & Bouché 1998 and *All. (Ga.) transmatica* Qiu & Bouché 1998, were described and added to the subgenus *Gatesona* [8]. The position of *Allolobophora chlorotica* (type species of *Allolobophora*) within a well-supported, independent clade from *All. (Ga.) chaetophora* [5] suggests that *Gatesona* could be erected as a proper genus. Molecular phylogenetic analysis of additional species (and subspecies) would contribute to a robust assessment of this hypothesis.

In this study, a systematic revision of *Allolobophora (Gatesona)* is proposed based on a molecular phylogenetic analysis including several species and subspecies. In addition, the phylogenetic link between *Allolobophora (Gatesona)*, *Helodrilus (Acystodrilus)* and *Avelona ligra* was tested by the inclusion of the later in the same phylogenetic data set.

## Materials and methods

### Specimens, sampling and morphological description

The specimens described in this study were collected in a sampling survey carried out in Southeastern France and Catalonia between October and November 2019, with additional sampling in Spring 2021. The list of species and the locations where they were collected are shown in Table 1.

Earthworms were obtained by soil digging and hand-sorting, rinsed with water and fixed in 70% in order to obtain relaxed (as opposed to excessively retracted) specimens. They were subsequently transferred to 100% ethanol to enable further molecular analyses. Species classification and morphological diagnoses were conducted using the same set of external and internal morphological characters used by Qiu & Bouché [8,9]. The following main external

**Table 1. Species sampled and sequenced in this study, with their sampling location and coordinates.**

| Taxon | Location | Coordinates |
|---|---|---|
| *Allolobophora (Gatesona) chaetophora chaetophora*[*] | L'Hospitalet du Larzac | 43.956, 3.218433 |
| *Allolobophora (Gatesona) chaetophora lablacherensis* Bouché 1972 | Lablachère, France | 44.433, 4.22643 |
| *Allolobophora (Gatesona) chaetophora serninensis* Bouché 1972 | Saint-Sernin-sur-Rance | 43.87487, 2.60635 |
| *Allolobophora (Gatesona) chaetophora rutena* Bouché 1972 | Rôdez, France | 44.415, 2.03943 |
| *Allolobophora (Gatesona) sausseyi* | Argelès-sur-Mer, France | 42.52361, 3.04193 |
| *Allolobophora (Gatesona) transmatica* | Agullana, Spain | 42.39626, 2.84851 |
| *Helodrilus (Acystodrilus) musicus* | Saint-Germain-de-Calberte, France | 44.21667, 3.79999 |
| *Avelona ligra* | Jargeau, France | Not available |
| *Allolobophora bartolii* Bouché 1970 | Jouques, France | 43.641, 5.63043 |

[*]Individuals from this species were newly sampled for morphological study but previously existing sequences were used for the phylogenetic analyses.

morphological characters were considered: mean length, mean number of segments, mean weight, pigmentation, type of prostomium, setal arrangement, position of papillae, position of first dorsal pore, nephridial pore arrangement, position and development of male pores, position and development of female pores, position of spermathecal pores, position of clitellum and position of tubercula pubertatis.

The main internal anatomical characters considered were as follows: position of oesophageal hearts, position and morphology of calciferous glands, position of crop, position of gizzard, type of typhlosole, shape of nephridial bladders, number and position of seminal vesicles, and number and position of spermathecae.

Holotypes of *Allolobophora (Gatesona) chaetophora* and its different subspecies (see Table 1), temporally available at the personal collection of Marcel Bouché, were studied and compared with the newly sampled material.

## DNA isolation, sequencing and molecular analyses

Total genomic DNA was extracted from ventral integument samples of approximately 5 x 5 mm by using the DNeasy Blood & Tissue Kit (Qiagen). For 8 species, regions of the nuclear 28S rRNA and mitochondrial 16S rRNA, tRNAs Ala, Leu and Ser, NADH dehydrogenase (*ND1*) and *COI* were amplified by polymerase chain reaction (PCR), with the primers and conditions described in Pérez-Losada *et al.* [10,11]. PCR products were purified and sequenced by the C.A.C.T.I Genomics service (University of Vigo).

DNA sequences obtained in this study are available in Genbank, under accession numbers MZ153163-MZ153169, MZ156573-MZ156588 and MZ235293-MZ235314.

Sequences reported by Domínguez *et al.* [5,12], Pérez-Losada *et al.* [10,11,13], Paoletti *et al.* [14] and de Sosa *et al.* [15], including representatives of most of the Lumbricidae genera and two members of the closest families (Hormogastridae and Criodrilidae), were downloaded from Genbank and used as a reference data set. The species included are listed in Supporting Information S1 File.

Sequences were aligned with MAFFT v.7 [16] with default settings and concatenated with BioEdit [17], resulting in a matrix of 3,201 bp. The best fitting evolutionary model for each

partition was selected with jModelTest v. 2.1.3 [18] by applying the Akaike information criterion (AIC [19]), and Bayesian information criterion (BIC [20]). GTR + I + G was selected as the best-fitting evolutionary model for COI, 28S and ND1, HKY+I+G was selected for 16S and HKY+G was selected for tRNAs.

Maximum Likelihood analysis was performed with RaxML-NG [21] as implemented in the CIPRES Science Gateway V. 3.3 [22], with 10 random starting trees and estimating the support for the resulting topologies with 1,000 rapid bootstrap replicates. Bayesian Inference of the phylogeny was estimated with MRBAYES v.3.1.2 [23] as implemented in the CIPRES Science Gateway V. 3.3. Parameters were set to 50 million generations and sampled every 5,000th generation (10,000 trees). Two independent runs each with four chains were performed and 20% of the trees were discarded as burn-in. The remaining trees were combined and summarized on a 50% majority-rule consensus tree. Clade support values over 70% and 90% (for Bootstrap and Posterior probability respectively) were considered high (see [15,24]).

Uncorrected average pairwise distances between the studied species for the molecular markers COI and 16S were calculated in MEGA X [25] in order to support their status as separate species and to study genetic distances within and between genera.

## Results and discussion

### Molecular phylogenetic analyses

Bayesian inference (Fig 1) and maximum likelihood inference phylogenetic trees recovered consistent topologies. *Avelona ligra* and representatives of *Helodrilus (Acystodrilus)* and *Allolobophora (Gatesona)* species formed a well-supported clade, isolated from species assigned to *Allolobophora* and *Helodrilus*. *Allolobophora (Gatesona)* and *Helodrilus (Acystodrilus)* appeared intermixed in a well-supported genus-level clade, hereafter referred to as *Gatesona*. Within *Gatesona*, representatives of the different subspecies of *All.* (*Ga.*) *chaetophora* were

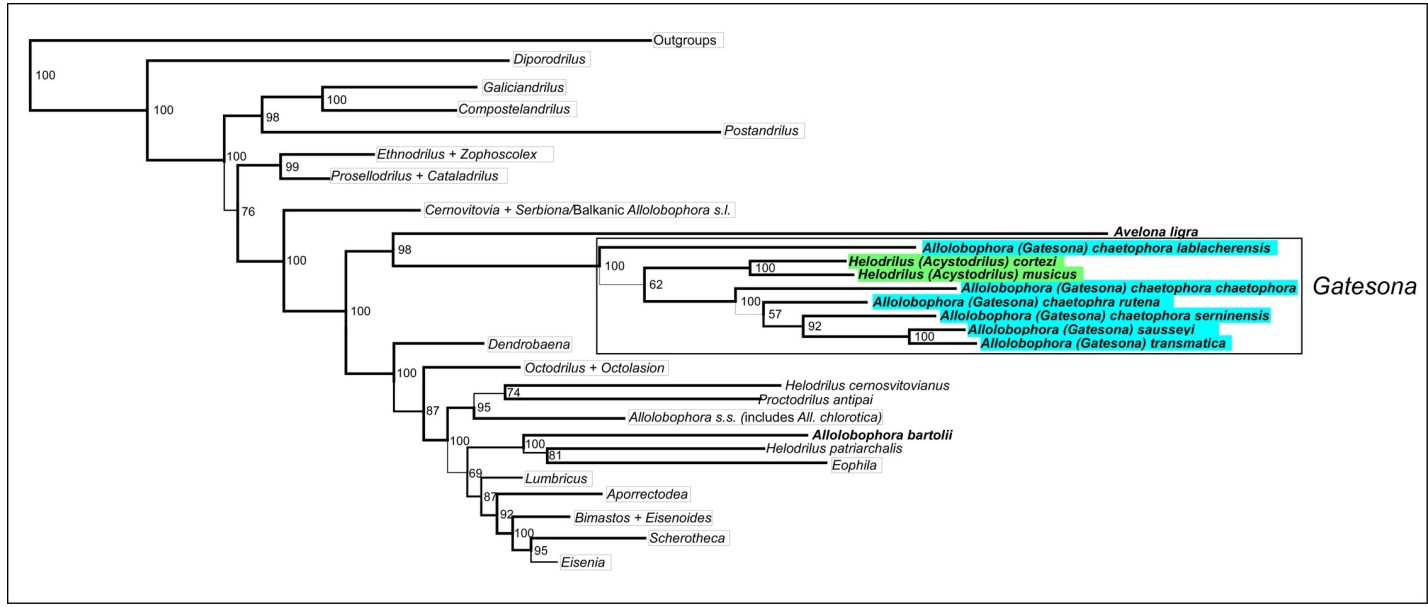

**Fig 1. Detail of the phylogenetic tree obtained by Bayesian phylogenetic analysis of the concatenated sequence of molecular markers *COI*–16S-tRNAs–*ND1*–28S (see full tree in S2 File).** *Allolobophora (Gatesona)* representatives are shaded in blue and *Helodrilus (Acystodrilus)* representatives are shaded in green. Representatives of the main Lumbricidae genera were collapsed and are shown in boxes. Posterior probability support values are shown besides the corresponding nodes.

separated by long branches. In addition, *Ga. sausseyi* and *Ga. transmatica* were nested within a clade including *All. (Ga.) chaetophora* and its different subspecies.

COI and 16s uncorrected average pairwise genetic (UAPG) distances between the species of the genera *Avelona*, *Gatesona* (comprising *All. (Gatesona)* and *He (Acystodrilus)*) and *Helodrilus (Helodrilus)* included in this study are shown in Table 2. UAPG distances between representatives of *Avelona*, *Gatesona* and *Helodrilus (Helodrilus)* were high, ranging from 15.2% to 21% for COI and from 11.0% to 17.5% for 16s. UAPG distances within the genus-level clades were lower, with values corresponding to *Gatesona* (8.1%-17.3% COI, 4.1%-13.2% 16s) being generally lower than distances within *Helodrilus (Helodrilus)* (17% COI, 11.9% 16s). UAPG distances between the different species and subspecies of *Gatesona* were high (see above) except for *All. (Ga.) sausseyi- All. (Ga.) transmatica* (8.1% COI, 4.1% 16s).

## Morphological analysis

External and internal morphological characters of the species and subspecies of *Allolobophora (Gatesona)* and related *Helodrilus (Acystodrilus)* species are shown in Table 3. Specimens were consistent with the holotypes and available descriptions [8,9,26] except for the aspects detailed below.

Specimens of *All. (Ga.) rutena* differed from the previous description in the number and position of spermathecal pores (2 pairs in 9/10, 10/11 vs 3 pairs in 9/10,10/11 and 11/12). However, the holotype of *All. (Ga.) rutena* showed 3 pairs of spermathecae on the right side but 2 pairs of spermathecae on the left side. Thus, this character appears to show variability even within the same individual.

**Table 2. Uncorrected average pairwise genetic (UAPG) distances between species of the genera *Avelona*, *Helodrilus (Acystodrilus)*, *Gatesona* and *Helodrilus (Helodrilus)* included in this study for the molecular markers COI (below the diagonal) and 16S (above the diagonal).**

| | *Av. ligra* | *He. (Ac.) cortezi* | *He. (Ac.) musicus* | *All. (Ga.) chaetophora* | *All. (Ga.) chaetophora lab.* | *All. (Ga.) chaetophora ser.* | *All. (Ga.) chaetophora rutena* | *All. (Ga.) sausseyi* | *All. (Ga.) transmatica* | *He. cernosvitovianus* | *He. patriarchalis* |
|---|---|---|---|---|---|---|---|---|---|---|---|
| *Av. ligra* | | 14.3 | 16.9 | 16.6 | 13.5 | 12.4 | 13.5 | 14.6 | 15.7 | 13.3 | 13.1 |
| *He. (Ac.) cortezi* | 19.6 | | 7.9 | 11.2 | 10.2 | 11.7 | 10.6 | 10.5 | 10.2 | 14.0 | 15.0 |
| *He. (Ac.) musicus* | 19.6 | 10.6 | | 12.5 | 12.5 | 13.1 | 12.0 | 9.7 | 11.6 | 14.8 | 16.3 |
| *All. (Ga.) chaetophora* | 15.2 | 16.0 | 16.0 | | 13.2 | 11.3 | 9.1 | 11.3 | 13.2 | 13.6 | 14.6 |
| *All. (Ga.) chaetophora lablacherensis* | * | * | * | * | | 10.1 | 9.0 | 9.7 | 10.4 | 11.4 | 14.2 |
| *All. (Ga.) chaetophora serninensis* | 19.0 | 16.7 | 16.2 | 14.5 | * | | 8.2 | 6.7 | 9.0 | 14.0 | 17.5 |
| *All. (Ga.) chaetophora rutena* | 17.8 | 9.1 | | 13.6 | * | 14.6 | | 7.5 | 7.9 | 11.0 | 13.5 |
| *All. (Ga.) sausseyi* | 18.3 | 17.3 | 16.5 | 14.4 | * | 14.5 | 14.1 | | 4.1 | 14.8 | 17.1 |
| *All. (Ga.) transmatica* | 18.5 | 16.7 | 16.3 | 13.2 | * | 12.9 | 14.1 | 8.1 | | 14.0 | 16.3 |
| *He. cernosvitovianus* | 20.5 | 19.0 | 18.8 | 17.5 | * | 19.5 | 17.3 | 19.8 | 19.1 | | 11.9 |
| *He. patriarchalis* | 19.1 | 18.6 | 18.8 | 17.7 | * | 20.1 | 17.2 | 18.2 | 18.2 | 17.0 | |

Red shading indicates higher values and green shading indicates lower values. The COI distance between *All. (Ga.) rutena* and *He. (Ac.) musicus* is not displayed as it was unusually low. Asterisks indicate COI sequences were not available for the species.

**Table 3. Main morphological characters of the species and subspecies of *Allolobophora* (*Gatesona*) and related *Helodrilus* (*Acystodrilus*) species under study.**

| Characters | All. (Ga.) chaetophora | All. (Ga.) chaetophora lablacherensis | All. (Ga.) chaetophora serninensis | All. (Ga.) chaetophora rutena | All. (Ga.) sausseyi | All. (Ga.) transmatica | Helodrilus (Ac.) cortezi* | Helodrilus (Ac.) musicus |
|---|---|---|---|---|---|---|---|---|
| Length | 68–85 mm | 88–100 mm | 70–99 mm | 98–115 mm | 86 mm | 66 mm | 152–192 mm | 127 mm |
| N. segments | 190 | 184–193 | 118–147 | 175–192 | 183 | 184 | 185–191 | 197 |
| Weight | 2.04–3.18 g | 1.64–1.72g | 1.085–1.675 g | 2.16–3.47 g | 1.25 g | 1.03 g | 1.10–1.36 g | 1.42 g |
| Pigmentation | Pigmentary dots | Pigmentary dots | Pigmentary dots | Pigmentary dots | Pigmentary dots | Pigmentary dots | Pigmentary dots | **Pigmentary dots** |
| Setal arrangement | 10:1.25:9.5:1:12.5 | 16:1:8.5:1:31 | 13.3:1:8.3:1:33 | 15:1:8.5:0.7:27.5 | 48:1:8:1:12 | 20:1:10:1:18 | 11.6:1.6:8.33:1:30 | 7:1:4:1:22 |
| First dorsal pore | 5/6 | 5/6 | 5/6 | 4/5 | 6/7 | (6/7)7/8 | 4/5 | 4/5 |
| Spermathecal pores | 9/10, 10/11 simple in C | 9/10, 10/11 simple in C | 9/10, 10/11 simple in C | **9/10, 10/11 simple in C** | 9/10, 10/11 simple in C | 9/10, 10/11 simple in C | 9/10, 10/11 simple in C | 9/10, 10/11 simple in C |
| Clitellum | 26–36 | 25–32 | 25–33 | 26–35 | 24–33 | 25–33 | 1/2 22–33 | 23–32 |
| Tubercula pubertatis | (29)30-36 | 27–30, strongly developed | 28-32(33) | 30–35 | 28–32 | 29–32 | 28–30 | 27-1/2 31 |
| Oesophageal hearts | 5–11 | 6–11 | 6–12 | 6–11 | 6–11 | 6–11 | 6–11 | 6–11 |
| Calciferous glands | 10–12, dilated in 10 | 10–12, dilated in 10 | 10–12, diverticles in 10 | 10–14, dilated in 10 | 10–14, diverticles in 10 | 10–12 | 10–14 | ½ 10–14, diverticles in 10 |
| Typhlosole | Bifid | Bifid | Bifid | Bifid | Bifid | Bifid | Bifid | Bifid |
| Nephridial bladders | Sigmoid, proclinate | Sigmoid, proclinate | Sigmoid, proclinate | Sigmoid, proclinate | Sigmoid, proclinate | Sigmoid, proclinate | Absent | Absent |
| Seminal vesicles | 9<10<11,12 | 10,11,12 | 9,10<11,12 | 9,10<11,12 | 9,10<11,12 | 9,11,12 | 9,10<11,12 | 9,10<11,12 |
| Spermathecae | Simple, globular in 9,10 | Simple, globular in 9,10 | Simple, globular in 9,10 | **Simple, globular in 9,10** | Simple, globular in 9,10 | Simple, globular in 9,10 | Simple, globular in 9,10 | Simple, globular in 9,10 |

Species indicated with *: information from Qiu and Bouché [9]. In bold, morphological characters for which differences were observed relative to the original description [26].

Specimens of *Helodrilus (Ac.) musicus* did not show red pigmentation as stated by [9], but displayed a faint reddish hue instead.

Nephridial pores were defined by [8,9,26] as irregularly distributed; however, newly captured material showed aligned nephridial pores.

## Systematic implications

The phylogenetic position of representatives of *Gatesona* as a well-supported clade unrelated to the generotype of *Allolobophora* (*All. chlorotica*) suggests it should be considered its own genus. The inclusion of *Helodrilus (Acystodrilus) cortezi* and *Helodrilus (Acystodrilus) musicus* within this clade suggest that they also belong to *Gatesona*. The latter two species and the other representatives of *Gatesona* are very similar in most external and internal morphological characters, except for the absence of nephridial vesicles. This character appears in several unrelated lumbricid lineages (Erséus, Csudzi pers. comm.) and is thus homoplasic: the same is observed for the posterior displacement of the male pore, which has stopped being regarded as a key taxonomic character in lumbricid systematics [12,27]. Regression of a complex morphological structure is easier in evolutionary terms than the evolution of such structures [28], so it is likely that loss of nephridial bladders happened repeatedly associated with high moisture environments [29].

Another species placed within *Helodrilus (Acystodrilus)* by Qiu and Bouché [9] and not included in this work, *Helodrilus (Acystodrilus) segalensis*, could be expected to belong to the same genus-level clade as *Helodrilus (Acystodrilus) cortezi* and *Ac. musicus*, due to their morphological similarity and geographic proximity (100–120 km); however, molecular phylogenetics analyses of this species is necessary to confirm this hypothesis. Furthermore, *Helodrilus putricola* and *tebra* have been found to belong to an unrelated clade (Csudzi pers. comm.*)* and *Helodrilus rifensis* is considered a synonym of *Eiseniella neapolitana* (Orley 1885) [30].

The name *Helodrilus* should be restricted to the clade containing *He. oculatus*, its generotype. However, *He. oculatus* was not included in this study, whereas only two Eastern European *Helodrilus (Helodrilus)* representatives (*Helodrilus patriarchalis* and *Helodrilus cernosvitovianus*) among its 18 species (http://taxo.drilobase.org/) were included. Thus, a well-supported systematic revision of *Helodrilus* should include a more comprehensive sampling of both groups in a molecular phylogenetic framework.

As *Avelona ligra* was recovered as a sister clade to *Gatesona*, it could be argued that it should also be included within the same genus. However, the branch lengths separating both taxa in the phylogenetic tree were indicative of separate genera. The 16s UAPG distances (which are less saturated at above-species level than COI distances) between *Avelona ligra* and representatives of *Gatesona* were significantly higher than those found within *Gatesona*, and were within the range observed between well-established earthworm genera (8.6%-20.01%, average 13.4%; Supporting Information S2 File). In addition, *Avelona ligra* (only known species within *Avelona*) possesses only two pairs of seminal vesicles in 11, 12 (vs 3 or 4 pairs in (9,10) 11, 12) and displays striking bioluminescence.

The phylogenetic relationships, branch lengths and UAPG distances between the different species and subspecies of *Gatesona* suggest that they should all be considered different species. The existing morphological differences between these taxa, of similar or higher magnitude than those used by Qiu and Bouché [8] to distinguish All. (*Ga.)* *sausseyi* and All. (*Ga.)* *transmatica*, support their status as proper species.

*All. (Ga.) argentatensis* could not be sampled for this study, but following the same criteria it is very likely that it constitutes an independent species from *All. (Ga.) chaetophora*.

## Proposed systematic changes

Phylum Annelida Lamarck, 1802
 Class Oligochaeta Grube, 1850/Clitellata Michaelsen, 1919
 Order Crassiclitellata Jamieson, 1988
 Family Lumbricidae Rafinesque-Schmaltz, 1815

## Genus *Gatesona* Qiu and Bouché 1998 stat. nov.

*Allolobophora (Gatesona)* Qiu and Bouché 1998

## Type species

*Allolobophora chaetophora* Bouché 1972

## Included species

*Gatesona chaetophora comb. nov.* (Bouché 1972)
 *Gatesona chaetophora argentatensis comb.nov.* (Bouché 1972)
 *Gatesona rutena comb.nov. stat. nov.* (Bouché, 1972)
 *Gatesona lablacherensis comb.nov.* (Bouché, 1972)

*Gatesona serninensis comb.nov. stat. nov.* (Bouché 1972)
*Gatesona sausseyi comb. nov.* (Qiu and Bouché 1998)
*Gatesona transmatica comb. nov.* (Qiu and Bouché 1998).
*Gatesona cortezi comb. nov.* (Qiu and Bouché 1998).
*Gatesona musica comb. nov.* (Qiu and Bouché 1998).

## Diagnosis

Mid-sized (66 to 190 mm) Lumbricidae, with fewer than 230 segments. Mid-segment pigmentary dots in cephalic (sometimes tail) segments. Numerous chaetophores in the shape of genital papillae between segments 9 and the end of the clitellum. Prostomium epylobous. Closely paired setae. First dorsal pore in 4/5 (3 species), 5/6 (3 species), 6/7 (1 species), 7/8 (1 species). Nephridial pores aligned. Male pores in ½ 15 with developed porophore. Spermathecal pores simple, in 9/10, 10/11 (exceptionally 9/10, 10/11, 11/12). Clitellum starts between segments 23 to 27. Calciferous gland variable, in 10-12(14) usually dilated or with diverticles in 10. Typhlosole bifid. Three to four pairs of seminal vesicles in (9, 10) 11, 12. Nephridial vesicles sigmoid, proclinate, sometimes absent (*Ga. cortezi*, *Ga. musica*).

## Remarks

*Gatesona chaetophora argentatensis* (Bouché 1972) remains a subspecies of *Gatesona chaetophora* owing to failure to include representatives in the integrative systematic revision. However, it is to be expected that *Ga. chaetophora argentatensis* could represent an independent species.

The name *Gatesona* was chosen over *Acystodrilus* for the genus encompassing species of both former subgenera for three main reasons: first, currently there are more species of the former *Allolobophora* (*Gatesona*) than of *Helodrilus* (*Acystodrilus*); second, the name *Acystodrilus* (meaning "without vesicle") does not represent the more common character state within the redefined genus *Gatesona*. The third reason is purely nomenclatural: the name *Gatesona* has priority over *Acystodrilus*, as far as [31] proposed *Gatesona* in page 201, whereas *Acystodrilus* was proposed in page 209.

## *Gatesona chaetophora* comb. nov stat. nov.

*Allolobophora chaetophora chaetophora* Bouché 1972

## Redescription

**Diagnosis.** Specimens of *Gatesona chaetophora* can be distinguished from other known species of *Gatesona* by the position of the clitellum in segments 26–36 and tubercula pubertatis in segments (29)30-36 (Table 3).

**Etymology.** The name *chaetophora* refers to the multiple chaetophores (glandular structures surrounding genital chaetae) this species possess.

**Material examined.** *Holotype.* FRANCE • Adult; Occitanie, Aveyron, L'Hospitalet du Larzac; 11/03/1968; leg. Marcel Bouché; Locality 1: 43.956 3.218433 Prairie; deposited in the collection of the Muséum National d'Histoire Naturelle de Paris (MNHN) (voucher: 87/550/0711).

*Additional material.* FRANCE • 2 specimens, adults; Occitanie, Aveyron, L'Hospitalet du Larzac; 11/02/2021; leg. Daniel F. Marchán; Locality 1: 43.956, 3.218433 Prairie; deposited in the CEFE earthworm collection (voucher: DFM-0676, DFM-0677).

**Morphological description.** *External morphology*. Body pigmentation faint brown. Beige to faint brown with pigmentary dots scattered along the mid-segment line in the cephalic region in fixed specimens (Fig 3A and 3E).

Mean length 76.5 mm (68–85 mm, n = 2 adults); body cylindrical in cross-section; mean number of segments 190 (n = 2 adults; 190 segments in the holotype). Mean weight (fixed specimens): 2.60 g (2.035–3.175 g, n = 2 adults). Prostomium epilobous, closed. Longitudinal furrows in segments 1 and 2. First dorsal pore at intersegmental furrow 5/6. Nephridial pores aligned in *B* close to *b*. Spermathecal pores at intersegmental furrows 9/10 and 10/11 in *c*. Male pores in segment 15, surrounded by a well-developed porophore. Female pores on the posterior part of segment 14, inconspicuous. Clitellum saddle-shaped in segments 26–36. Tubercula pubertatis in segments (29)30-36. Chaetae small and closely paired, with interchaetal ratio *aa*: 10, *ab*: 1.25, *bc*: 9.5, *cd*: 1, *dd*: 12.5 at segment 40. Chaetophores/genital papillae in segments 11, (14, 16, 17, 18) 20, 25–28, (29), 32–36 with those in 11, 20, 25–28 and 35,36 especially well developed and constant.

*Internal anatomy*. Septa 5/6-9/10 strongly thickened. Hearts in segments 5–11, oesophageal. Calciferous glands in segments 10–12, dilated in segment 10. Crop in segments 15–16, gizzard in segments 17–18. Typhlosole bifid. Male sexual system holandric, testes and funnels (not enclosed in testes sacs, but with sperm present) located ventrally in segments 10 and 11. Four pairs of reniform seminal vesicles in segments 9, 10, 11 and 12, with the latter two pairs being larger. Ovaries and female funnels in segment 13, ovarian receptacles (ovisacs) in segment 14. Two pairs of globular spermathecae in segments 9 and 10. Nephridial bladders sigmoid, proclinate in anterior and posterior segments (Fig 3I).

**Distribution and ecology:** *Gatesona chaetophora* is known from the region of Larzac in the southern Massif Central, France. This species was found at an altitude of 710 meters in a prairie on silty soil, with close to neutral pH (7.4), 2.83% organic matter and 0.308% nitrogen content [26].

## *Gatesona rutena* comb. nov stat. nov.

*Allolobophora chaetophora rutena* Bouché 1972

## Redescription

**Diagnosis.** Specimens of *Gatesona rutena* can be distinguished from other known species of *Gatesona* by the position of the first dorsal pore in 4/5, the position of the clitellum in segments 26–35 and tubercula pubertatis in segments 30–35, calciferous glands in 10–14 and a higher mean body weight (Table 3).

**Etymology.** *Gatesona rutena* was named after the Gallic tribes (the Ruten) that inhabited French lands before Roman invasion (see Bouché [26] page 5).

**Material examined.** *Holotype*. FRANCE • Adult; Occitanie, Aveyron, Rodez; 17/04/1968; leg. Marcel Bouché; Locality 1: 44.415 2.039433 Pasture; deposited in the collection MNHN (voucher: 91/619/0712).

*Additional material*. FRANCE • 7 specimens, 4 adults, 3 juveniles; Occitanie, Aveyron, Rodez; 10/11/2019; leg. Daniel F. Marchán; Locality 1: 44.416565 2.041011 Oak forest; deposited in the CEFE earthworm collection (voucher: 70257).

**Morphological description.** *External morphology*. Body pigmentation very faint brown, white-creamy spots can be observed through the body wall (Fig 2A). White-beige with pigmentary dots scattered along the mid-segment line in the cephalic (sometimes tail) region in fixed specimens (Fig 3B and 3F).

Mean length 106.5 mm (98–115 mm, n = 2 adults); body cylindrical in cross-section; mean number of segments 183 (175–192, n = 2 adults; 192 segments in the holotype). Mean weight (fixed specimens): 2.82 g (2.16–3.47 g, n = 2 adults). Prostomium epilobous, closed. Longitudinal furrows in segments 1 and 2. First dorsal pore at intersegmental furrow 4/5. Nephridial pores aligned in *B* close to *b*. Spermathecal pores at intersegmental furrows 9/10 and 10/11 (plus 11/12 according to Bouché 1972 page 439) in *c*. Male pores in segment 15, surrounded by a well-developed porophore. Female pores on the posterior part of segment 14, inconspicuous. Clitellum saddle-shaped in segments 26–35. Tubercula pubertatis in segments 30–35. Chaetae small and closely paired, with interchaetal ratio *aa*: 15, *ab*: 1, *bc*: 8.5, *cd*: 0.7, *dd*: 27.5 at segment 40. Chaetophores/genital papillae in segments 12, 16, 17, 27, 24–36.

*Internal anatomy*. Septum 4/5 thickened, septa 5/6-8/9 strongly thickened. Hearts in segments 6–11, oesophageal. Calciferous glands in segments 10–14, dilated in segment 10. Crop in segments 15–16, gizzard in segments 17–18. Typhlosole bifid. Male sexual system holandric, testes and funnels (not enclosed in testes sacs, but with sperm present) located ventrally in segments 10 and 11. Four pairs of reniform seminal vesicles in segments 9, 10, 11 and 12, with the latter two pairs being larger. Ovaries and female funnels in segment 13, ovarian receptacles (ovisacs) in segment 14. Two pairs of globular spermathecae in segments 9 and 10 (an additional spermathecae can appear in 11 in one or both sides of the body). Nephridial bladders sigmoid, proclinate in anterior and posterior segments (Fig 3J).

**Distribution and ecology:** *Gatesona rutena* is known from the region of Rodez, in the western Massif Central, France. This species was found at an altitude of 360 meters in a prairie/oak forest transition on silty soil, with close to neutral pH (7.2), 4.79% organic matter and 0.414% nitrogen content [26].

## *Gatesona lablacherensis* comb. nov stat. nov.

*Allolobophora chaetophora lablacherensis* Bouché 1972

### Redescription

**Diagnosis.** Specimens of *Gatesona lablacherensis* can be distinguished from other known species of *Gatesona* by the position of the clitellum in segments 25–32 and tubercula pubertatis in segments 27–30, as well as by possessing three pairs of seminal vesicles (in segments 10, 11 and 12) rather than four (Table 3).

**Etymology.** The name *lablacherensis* refer to Lablachère, the region of France the species is known to inhabit.

*Holotype*. FRANCE • Adult; Rhône-Alps, Ardèche, Lablachère; 20/11/1968; leg. Marcel Bouché; Locality 1: 44.433, 4.22643 Oak forest over rocky outcrops; deposited in MNHN (voucher: 90/1056/0716).

*Additional material*. FRANCE • 12 specimens, adults; Rhône-Alps, Ardèche, Lablachère; 6/11/2019; leg. Daniel F. Marchán; Locality 1: 44.433, 4.22643 Oak forest over rocky outcrops; deposited in the CEFE earthworm collection (voucher: 70256).

**Morphological description.** *External morphology*. Body pigmentation absent, very conspicuous white-creamy spots can be observed through the body wall (Fig 2B). White-beige with pigmentary dots scattered along the mid-segment line in the cephalic region in fixed specimens (Fig 3C and 3G).

Mean length 94 mm (88–100 mm, n = 2 adults); body cylindrical in cross-section; mean number of segments 189 (184–193, n = 2 adults; 193 segments in the holotype). Mean weight (fixed specimens): 1.68 g (1.64–1.72 g, n = 2 adults). Prostomium epilobous, closed. Longitudinal furrows in segments 1 and 2. First dorsal pore at intersegmental furrow 5/6. Nephridial

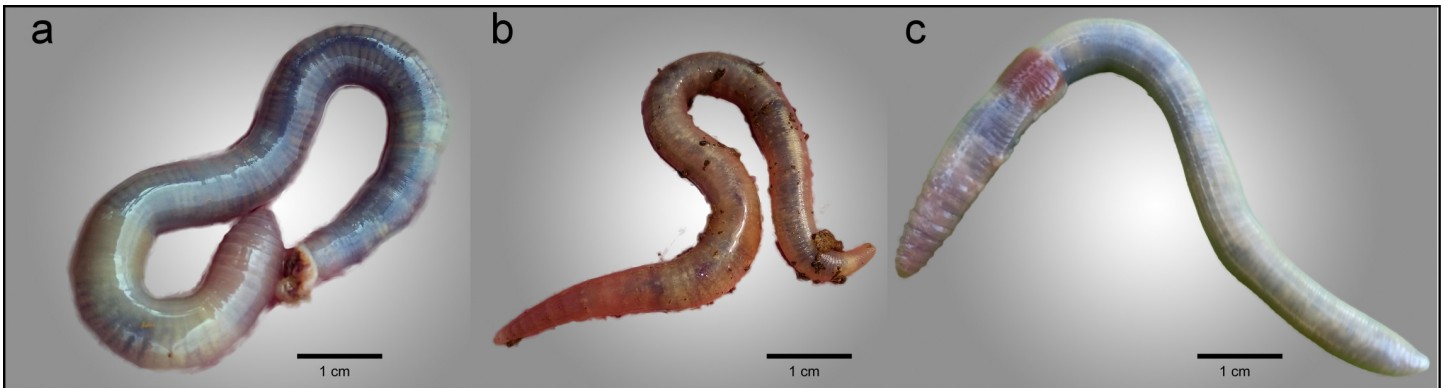

**Fig 2.** External morphology of living specimens of a) *Gatesona rutena* b) *Gatesona lablacherensis* c) *Gatesona serninensis*.

**Fig 3.** External and internal morphology of fixed specimens of *Gatesona chaetophora* (a, e, i) *Gatesona rutena* (b, f, j) *Gatesona lablacherensis* (c, g, k) and *Gatesona serninensis* (d, h, l). The insets show a detailed view of the pigmentary dots characteristic of *Gatesona* species (e, f, g, h), and their nephridial bladders (i, j, k, l). Abbreviations: cl. = clitellum, m. p. = male pore, t. p. = tubercula pubertatis.

pores aligned in *B* close to *b*. Spermathecal pores at intersegmental furrows 9/10 and 10/11 in *c*. Male pores in segment 15, surrounded by a well-developed porophore. Female pores on the posterior part of segment 14, inconspicuous. Clitellum saddle-shaped in segments 25–32. Tubercula pubertatis in segments 27–30 strongly developed as downward projections of the body wall. Chaetae small and closely paired, with interchaetal ratio *aa*: 16, *ab*: 1, *bc*: 8.5, *cd*: 1, *dd*: 31 at segment 40. Chaetophores/genital papillae in segments 13, 14, 16, 17, 19, 23–33.

*Internal anatomy*. Septum 4/5 thickened, septa 5/6-8/9 strongly thickened. Hearts in segments 6–11, oesophageal. Calciferous glands in segments 10–12, dilated in segment 10. Crop in segments 15–16, gizzard in segments 17–18. Typhlosole bifid. Male sexual system holandric, testes and funnels (not enclosed in testes sacs, but with sperm present) located ventrally in segments 10 and 11. Three pairs of reniform seminal vesicles in segments 10, 11 and 12, with the latter two pairs being larger. Ovaries and female funnels in segment 13, ovarian receptacles (ovisacs) in segment 14. Two pairs of globular spermathecae in segments 9 and 10. Nephridial bladders sac-like or J shaped in the first segments, which gradually change to sigmoid, proclinate (Fig 3K).

**Distribution and ecology:** *Gatesona lablacherensis* is known from the region of Lablachère, in the eastern Massif Central, France. This species has been found in an oak forest, and thus may prefer natural habitats.

This species was found at an altitude of 180 meters in a prairie/oak forest transition on silty soil over rocks, with close to neutral pH (7.6), 1.9% organic matter and 0.166% nitrogen content [26].

## *Gatesona serninensis* comb. nov stat. nov.

*Allolobophora chaetophora serninensis* Bouché 1972

### Redescription

**Diagnosis.** Specimens of *Gatesona serninensis* can be distinguished from other known species of *Gatesona* by the position of the clitellum in segments 25–33 and tubercula pubertatis in segments 28-32(33), as well as by the significantly lower number of segments (Table 3). *Gatesona serninensis* is similar to *Gatesona transmatica* in the position of tubercula pubertatis (28-32(33) vs 29–32) and they share the same position of the clitellum, but they are easily differentiated by the position of the first dorsal pore (5/6 vs (6/7) 7/8) and the number and position of seminal vesicles (4 in 9, 10, 11, 12 vs 3 in 9, 11, 12).

**Etymology.** The name *serninensis* refer to Saint-Sernine-sur-Rance, the town in the vicinities of whichr the species are known to inhabit.

*Holotype*. FRANCE • Adult; Occitanie, Aveyron, Saint-Sernin-sur-Rance; 26/11/1968; leg. Marcel Bouché; Locality 1: 43.875 2.606433 Prairie; deposited in MNHN (voucher 88/1100/0714)

*Additional material*. FRANCE • 12 specimens, 8 adults, 4 juveniles; Occitanie, Aveyron, Saint-Sernin-sur-Rance; 23/04/2021; leg. Daniel F. Marchán; Locality 1: 43.956, 3.218433 Prairie; deposited in the CEFE earthworm collection (voucher 70258).

**Morphological description.** *External morphology*. Body pigmentation absent, conspicuous white-creamy spots can be observed through the body wall. White-beige with pigmentary dots scattered along the mid-segment line in the cephalic (sometimes caudal) region in fixed specimens (Fig 3D and 3H).

Mean length 85 mm (70–99 mm n = 6 adults); body cylindrical in cross-section; mean number of segments 128 (118–147, n = 8 adults; 121 segments in the holotype). Mean weight (fixed specimens): 1.32 g (1.085–1.675 g, n = 4 adults). Prostomium epilobous, closed.

Longitudinal furrows in segments 1 and 2. First dorsal pore at intersegmental furrow 5/6. Nephridial pores aligned in *cd*. Spermathecal pores at intersegmental furrows 9/10 and 10/11 in *c*. Male pores in segment 15, surrounded by a well-developed porophore. Female pores on the posterior part of segment 14, inconspicuous. Clitellum saddle-shaped in segments 25–33. Tubercula pubertatis in segments 28–32 (33). Chaetae small and closely paired, with interchaetal ratio *aa*: 13.3, *ab*: 1, *bc*: 8.3, *cd*: 1, *dd*: 33 at segment 40. Chaetophores/genital papillae in segments 11, 16, 17, 19–33, those in 11, 16, 32 and 33 especially well developed and constant.

*Internal anatomy*. Septa 5/6-9/10 strongly thickened. Hearts in segments 6–12, oesophageal. Calciferous glands in segments 10–12, with diverticles in segment 10. Crop in segments 15–16, gizzard in segments 17–18. Typhlosole bifid. Male sexual system holandric, testes and funnels (not enclosed in testes sacs, but with sperm present) located ventrally in segments 10 and 11. Four pairs of reniform seminal vesicles in segments 9, 10, 11 and 12, with the latter two pairs being larger. Ovaries and female funnels in segment 13, ovarian receptacles (ovisacs) in segment 14. Two pairs of globular spermathecae in segments 9 and 10. Nephridial bladders J shaped in the first segments, which gradually change to sigmoid, proclinate (Fig 3L).

**Distribution and ecology:** *Gatesona serninensis* is known from the region of Saint-Sernin-sur-Rance, in the southern Massif Central, France. This species has been found at an altitude of 350 meters in the transition between an oak-chestnut forest and a prairie on sandy soil over rocks, with acid pH (5.6), 3.78% organic matter and 0.350% nitrogen content [26].

## Conclusions

Integrative systematics supported the status of the previously described subgenus *Allolobophora (Gatesona)* as a genus, including some representatives of *Helodrilus (Acystodrilus)*. Additionally, a clade comprising *Gatesona* and *Avelona* was recovered by molecular phylogenetics, an early branching clade with most of its species being endemic to the Massif Central. Morphological and molecular evidence indicated most of the previously described subspecies of *Gatesona chaetophora* should actually be regarded as proper species. The presence of this diverse assemblage of endemic earthworms, with several taxa remaining to be studied by molecular methods, highlights the interest of this region of Southern France for further sampling effort across their range.

The phylogenetic distinctness of representatives of *Helodrilus (Helodrilus)* and *Helodrilus (Acystodrilus)* indicate that a more comprehensive sampling of species (including *He. oculatus*) is required in order to establish a solid systematic revision of the elusive genus.

## Supporting information

**S1 File. Species included in the reference dataset.**
(DOCX)

**S2 File. Phylogenetic tree obtained by Bayesian phylogenetic analysis of the concatenated sequence of molecular markers *COI*–16S-tRNAs–*ND1*–28S.**
(DOCX)

**S3 File. Uncorrected average pairwise (UAPG) distances for the marker 16S between representatives of different well-established genera of Lumbricidae.**
(DOCX)

## Acknowledgments

We are indebted to Marcel Koken who kindly provided the specimens of *Avelona ligra*.

We are also thankful to Marcel Bouché for making the holotypes available for morphological study.

## Author Contributions

**Conceptualization:** Daniel F. Marchán, Jorge Domínguez.

**Data curation:** Daniel F. Marchán, Sergio Jiménez.

**Formal analysis:** Daniel F. Marchán.

**Funding acquisition:** Daniel F. Marchán, Jorge Domínguez.

**Investigation:** Daniel F. Marchán, Sergio Jiménez, Thibaud Decaëns.

**Methodology:** Daniel F. Marchán.

**Project administration:** Jorge Domínguez.

**Resources:** Thibaud Decaëns, Jorge Domínguez.

**Supervision:** Daniel F. Marchán, Thibaud Decaëns, Jorge Domínguez.

**Validation:** Daniel F. Marchán.

**Visualization:** Daniel F. Marchán.

**Writing – original draft:** Daniel F. Marchán.

**Writing – review & editing:** Daniel F. Marchán, Jorge Domínguez.

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
