## [Decision Letter · Decision Letter 0]

15 Apr 2021

PONE-D-21-07434

Systematic revision of Gatesona (Crassiclitellata, Lumbricidae), an endemic earthworm genus from the Massif Central (France)

PLOS ONE

Dear Dr. Marchán,

Thank you for submitting your manuscript to PLOS ONE. After careful consideration, we feel that it has merit but does not fully meet PLOS ONE’s publication criteria as it currently stands. Therefore, we invite you to submit a revised version of the manuscript that addresses the points raised during the review process.

We look forward to receiving your revised manuscript.

Kind regards,

Michael Schubert

Academic Editor

PLOS ONE

Journal Requirements:

Reviewers' comments:

Reviewer's Responses to Questions

**Comments to the Author**

1. Is the manuscript technically sound, and do the data support the conclusions?

Reviewer #1: Yes

Reviewer #2: Yes

2. Has the statistical analysis been performed appropriately and rigorously? 

Reviewer #1: Yes

Reviewer #2: Yes

3. Have the authors made all data underlying the findings in their manuscript fully available?

Reviewer #1: Yes

Reviewer #2: Yes

4. Is the manuscript presented in an intelligible fashion and written in standard English?

Reviewer #1: Yes

Reviewer #2: No

5. Review Comments to the Author

Reviewer #1: Authors present a very interesting paper about the identity of several Lumbricidae species and subspecies that are placed in a new genus. These new arrangements are due to the use of an integrative approach (molecular genetics, phylogenetic analysis and morphological revisions). English is ok, as well as molecular and phylogenetic methods and results.

There are, however, some modifications that should be done (mainly in regard to classical nomenclatural arrangements) before the paper could be accepted. The paper, thus, is ACCEPTED WITH MINOR CHANGES.

I directly wrote several comments and suggestions over the main text; following are some of my main points:

1) In the title it is mentioned the word Crassiclitellata; in the classification of the new genus Gatesona, however, this word is not included. Thus, include in the supraorder classification this word with its hierarchical associated level.

2) There are several mistakes in regard of the “types” of the two re-described species. Please attend comments of the text.

3) As authors do not reviewed types of these species, maybe the name (from an already described species) in fact does not correspond to the collected and molecular sequenced material. I strongly suggest contact Dr. Bouché in order to obtain in loan the types (or paratypes) to confirm that the morphology of Gatesona chaetophora rutena and Gatesona chaetophora lablacherensis really correspond to the species you collected. Even more because there is some variation that could be an indication of different morphological species (e.g. the number of spermathecae).

4) In benefit of readers, please explain in the discussion why the genus was named as Gatesona and not Acystodrilus.

Reviewer #2: There are a few small corrections of grammar and content in the MS that I uploaded with this review. The revision of the taxonomy has to be modified to make it clear that G. chaetophora is the type species, and in particular, that G. chaetophora chaetophora is the type species. The subspecies G. c. argentatus could be tentatively excluded. The remaining species are "other species" under a heading.

6. PLOS authors have the option to publish the peer review history of their article (what does this mean?). If published, this will include your full peer review and any attached files.

Reviewer #1: No

Reviewer #2: No

---

## [Author Response · Author response to Decision Letter 0]

20 Jun 2021

Reviewer #1: Authors present a very interesting paper about the identity of several Lumbricidae species and subspecies that are placed in a new genus. These new arrangements are due to the use of an integrative approach (molecular genetics, phylogenetic analysis and morphological revisions). English is ok, as well as molecular and phylogenetic methods and results.

We want to thank the reviewer for his/her interest on our manuscript and his/her constructive suggestions in order to improve it. All of them have been considered and included.

There are, however, some modifications that should be done (mainly in regard to classical nomenclatural arrangements) before the paper could be accepted. The paper, thus, is ACCEPTED WITH MINOR CHANGES.

I directly wrote several comments and suggestions over the main text; following are some of my main points:

1) In the title it is mentioned the word Crassiclitellata; in the classification of the new genus Gatesona, however, this word is not included. Thus, include in the supraorder classification this word with its hierarchical associated level.

We have included Crassiclitellata in the taxonomy section.

2) There are several mistakes in regard of the “types” of the two re-described species. Please attend comments of the text.

We have corrected the taxonomic errors regarding the types.

3) As authors do not reviewed types of these species, maybe the name (from an already described species) in fact does not correspond to the collected and molecular sequenced material. I strongly suggest contact Dr. Bouché in order to obtain in loan the types (or paratypes) to confirm that the morphology of Gatesona chaetophora rutena and Gatesona chaetophora lablacherensis really correspond to the species you collected. Even more because there is some variation that could be an indication of different morphological species (e.g. the number of spermathecae).

We have contacted Marcel Bouché and revised the holotypes for lablacherensis, rutena but also serninensis and chaetophora. Thanks to that, we can state with certainty that the (morpho-molecularly) studied material is morphologically compatible with the types of the original taxa. The morphological differences initially detected could be actually due to faulty/incomplete descriptions (in the end, it was done through tables and not through an orthodox text); when comparing with the types, the differences were non-existant or comprised within intra-individual variation (Gatesona rutena holotype possessed three spermathecae on one side but only two on the other).

4) In benefit of readers, please explain in the discussion why the genus was named as Gatesona and not Acystodrilus.

We have included a sentence explaining that point in the “Remarks” section of the Systematic changes to the genus Gatesona.

Reviewer #2: There are a few small corrections of grammar and content in the MS that I uploaded with this review. The revision of the taxonomy has to be modified to make it clear that G. chaetophora is the type species, and in particular, that G. chaetophora chaetophora is the type species. The subspecies G. c. argentatus could be tentatively excluded. The remaining species are "other species" under a heading.

We are thankful for the constructive revision and suggestions for improvement of our manuscript. As seen in the document with tracked changes, we have aknowledged and included all the suggested changes.

Most importantly, we contacted Marcel Bouché and revised the holotypes for lablacherensis, rutena but also serninensis and chaetophora. This allowed to check for morphological differences, actually ruling out most of them (holotypes were more consistent with our material tan suggested in the incomplete, informal descriptions provided in Bouché 1972). In addition, we sampled additional material of Gatesona serninensis, which allowed us to provide a formal redescription.

We have decided to mantain argentatensis within Gatesona chaetophora as it clearly belongs to the genus but we lack enough information to robustly elevate it to its own species.

---

## [Decision Letter · Decision Letter 1]

26 Jul 2021

PONE-D-21-07434R1

Systematic revision of Gatesona (Crassiclitellata, Lumbricidae), an endemic earthworm genus from the Massif Central (France)

PLOS ONE

Dear Dr. Marchán,

Thank you for submitting your manuscript to PLOS ONE. After careful consideration, we feel that it has merit but does not fully meet PLOS ONE’s publication criteria as it currently stands. Therefore, we invite you to submit a revised version of the manuscript that addresses the points raised during the review process.

We look forward to receiving your revised manuscript.

Kind regards,

Michael Schubert

Academic Editor

PLOS ONE

Journal Requirements:

Reviewers' comments:

Reviewer's Responses to Questions

**Comments to the Author**

1. If the authors have adequately addressed your comments raised in a previous round of review and you feel that this manuscript is now acceptable for publication, you may indicate that here to bypass the “Comments to the Author” section, enter your conflict of interest statement in the “Confidential to Editor” section, and submit your "Accept" recommendation.

Reviewer #1: All comments have been addressed

2. Is the manuscript technically sound, and do the data support the conclusions?

Reviewer #1: Yes

3. Has the statistical analysis been performed appropriately and rigorously? 

Reviewer #1: Yes

4. Have the authors made all data underlying the findings in their manuscript fully available?

Reviewer #1: Yes

5. Is the manuscript presented in an intelligible fashion and written in standard English?

Reviewer #1: Yes

6. Review Comments to the Author

Reviewer #1: The paper is now ready for its publication. I'm, however, sending in the 2nd round reviewed paper very few suggestions and changes (mainly of style) to improve the paper. These suggestions include change of a word, italics, improve of figure 3 legend, a doubt about the identity of a personal communication and (and this is up to the author to include or not) the addition of a nomenclatural comment.

7. PLOS authors have the option to publish the peer review history of their article (what does this mean?). If published, this will include your full peer review and any attached files.

Reviewer #1: No

---

## [Author Response · Author response to Decision Letter 1]

26 Jul 2021

We have accepted all the proposed changes

---

## [Editor Report · Decision Letter 2]

28 Jul 2021

Systematic revision of Gatesona (Crassiclitellata, Lumbricidae), an endemic earthworm genus from the Massif Central (France)

PONE-D-21-07434R2

Dear Dr. Marchán,

We’re pleased to inform you that your manuscript has been judged scientifically suitable for publication and will be formally accepted for publication once it meets all outstanding technical requirements.

Kind regards,

Michael Schubert

Academic Editor

PLOS ONE

---

## [Editor Report · Acceptance letter]

25 Aug 2021

PONE-D-21-07434R2 

Systematic revision of *Gatesona* (Crassiclitellata, Lumbricidae), an endemic earthworm genus from the Massif Central (France) 

Dear Dr. Marchán:

I'm pleased to inform you that your manuscript has been deemed suitable for publication in PLOS ONE. Congratulations! Your manuscript is now with our production department. 

Kind regards, 

on behalf of

Dr. Michael Schubert 

Academic Editor

PLOS ONE